# Is resilience linked to stress response among anesthesia professionals? A prospective simulation-based study

Yoann Zafiriou [1]*, Jean-Noël Evain [2], Barthélémy Bertrand [3], Guillaume Archer[4], Joris Botton[4], Julien Crozet[4], Ismaël Khediri[4], Mathieu Lefevre[4], Camille Marcel[4], Charline Sery[4], Anne Ego[5], Pierre Albaladejo[2], Julien Picard[2]

1 Univ. Grenoble Alpes, CNRS, CHU Grenoble Alpes, Institute of Nursing Education, Grenoble INP(Institute of Engineering, Univ. Grenoble Alpes), TIMC-IMAG, Grenoble, France, 2 Univ. Grenoble Alpes, CNRS, CHU Grenoble Alpes, Department of Anesthesiology and Intensive Care, Grenoble INP(Institute of Engineering, Univ. Grenoble Alpes), TIMC-IMAG, Grenoble, France, 3 Univ. Grenoble Alpes, CHU Grenoble Alpes, Department of Anesthesiology and Intensive Care, Grenoble, France, 4 Univ. Grenoble Alpes, CHU Grenoble Alpes, School of Nurse Anesthetist Training, Grenoble, France, 5 Univ. Grenoble Alpes, Inserm, CHU Grenoble Alpes, CIC, Grenoble, France

* yzafiriou@chu-grenoble.fr

## Abstract

### Objectives

Anesthesiology is a stressful specialty characterized by high rates of burnout among its practitioners. Resilience, defined as a dynamic process of positive adaptation in the face of adversity, may serve as a protective factor. However, the relationship between resilience and the acute stress response remains poorly understood. This study aimed to test the hypothesis that higher resilience levels would be associated with attenuated physiological and psychological stress responses during a standard-ized anesthetic crisis simulation.

### Methods

Participants included anesthesiologists, anesthesiology residents, nurse anesthetists, and nurse anesthetist students. Resilience was measured using the Connor–David-son Resilience Scale (CD-RISC 10, range 0–40). Participants underwent a high-fidelity pediatric laryngospasm simulation. Physiological (heart rate variability) and psychological (self-reported stress) responses to acute stress were analyzed.

### Results

Thirty-four professionals were included. The median resilience score was 29 [27 –32], ranging from 18 to 34. No significant association was found between resilience and physiological (p=0.085) or psychological (p=0.621) responses to acute stress. Resilience was not related to age, years of experience, or professional role.

**Data availability statement:** All relevant data are within the manuscript and its Supporting Information files.

**Funding:** The author(s) received no specific funding for this work.

**Competing interests:** The authors have declared that no competing interests exist.

## Conclusions

Resilience scores were comparable to those reported in other studies. No significant association was observed between resilience and the acute stress response. These findings are consistent with theoretical frameworks that conceptualize resilience as a long-term adaptation process rather than an immediate reaction to stress. Direct measurement of resilience may therefore be more appropriate than relying on observable stress responses to identify professionals at psychosocial risk. Future studies using longitudinal designs and larger samples are needed to clarify these relationships.

## Introduction

Stress is a complex concept encompassing physiological, psychological, and behavioral reactions to a perceived threat. The transactional model of stress developed by Lazarus and Folkman posits that stress depends on each individual's cognitive appraisal of the situation and on the resources available to cope with it. This explains why individuals react differently to the same stressor: some will appraise a given situation as a threat, while others will perceive it as a challenge. [1] It is important to distinguish between acute stress (an immediate response to a stressor), perceived stress (a cognitive appraisal of stress), and chronic stress, which can lead to burnout. [2,3] From a physiological standpoint, the acute stress response activates the sympathetic nervous system, resulting in a decrease in heart rate variability (decreased SDNN). [4] Repeated or chronic stress exposure can lead to allostatic load, with long-term effects on metabolism, the cardiovascular system, and mental health. [5,6] Anesthesiology is recognized as a profession exposed to stress due to the nature of its work: critical care, rapid decision-making under uncertainty. [2,7,8] Recent studies show that 67.7% of American anesthesiologists were at high risk for burnout, with elevated rates among residents. [9,10] Stress can impair clinical performance [11,12] by altering cognitive processes [13] and is a source of professional burnout. Identifying professionals at risk of developing stress-related disorders is therefore a genuine challenge.

To cope with stress, several studies highlight the importance of developing a high level of resilience. [14,15] In psychology, the concept of resilience refers to the ability to face adversity and to overcome psychological trauma. [16,17] There are many definitions of resilience, but certain common terms stand out, such as bouncing back, adapting, and adjusting, describing resilience as a dynamic process that evolves over time. [18] Resilience does not imply the absence of distress during a stressful event; rather, it reflects the ability to recover and maintain psychological well-being over time. [19] Within the transactional model of stress, resilience can be viewed as a personal resource that influences the cognitive appraisal of the stressful situation. However, the relationship between resilience as a personality trait and the immediate stress response during acute clinical events remains poorly understood. While resilience is theoretically linked to adaptation, it is unclear whether greater resilience

translates into attenuated physiological or psychological responses during an acute stressful event, or whether its protective effects are primarily exerted through longer-term recovery processes. The literature suggests that junior anesthesiologists are more stressed than their more senior counterparts [8] and that resilience may vary with experience. [20] High-fidelity simulation allows for controlled conditions necessary to induce a stress response. [21] Pediatric laryngospasm is a common complication in anesthesia and lends itself well to a stressful simulation. [22]

Based on the transactional model of stress and resilience as a dynamic adaptation process, we formulated four hypotheses: (H1) higher resilience scores would be associated with an attenuated physiological stress response (higher SDNN); (H2) higher resilience scores would be associated with lower perceived acute stress; (H3) higher resilience scores would be inversely associated with perceived stress over the past month (PSS-10); (H4) resilience would be associated with age and professional experience. The primary objective of this exploratory study was to assess the association between resilience and physiological stress responses during a standardized anesthetic crisis simulation. Secondary objectives were to examine associations between resilience and acute perceived stress, perceived stress over the past month (PSS-10), age, and professional experience.

## Methods

### Study design

This was a prospective, single-center, observational descriptive study using high-fidelity simulation. The protocol was preregistered on ClinicalTrials.gov (NCT05073445) on June 18, 2021. Approval was obtained from the Ethics Committee of the French Society of Anesthesia and Intensive Care Medicine (IRB 00010254–2021–097) on May 8, 2021. All participants received written information about the study by email prior to inclusion and provided oral informed consent before participation. Because the study was classified as non-interventional and exempt from the French regulatory framework governing research involving human participants, the ethics committee did not require written informed consent. The study was conducted in accordance with the Declaration of Helsinki. The study was conducted at the healthcare professional training institute of the School of Nurse Anesthetists at Grenoble University Hospital, France. All participants were healthcare professionals or student volunteers from Grenoble Alpes University Hospital. The sample was selected by convenience among volunteer healthcare professionals participating in scheduled simulation sessions. Participants were recruited between June 18 and October 13, 2021. All participants received prior written information and provided verbal consent before participation, in accordance with ethics committee approval. Exclusion criteria were as follows: cardiac arrhythmias, endocrine disorders, hypertension, current pregnancy or breastfeeding, use of anti-inflammatory or psychotropic medications that could interfere with heart rate, and cessation of professional practice in anesthesia for more than 2 years.

The study timeline and measurements are presented in Table 1.

**Table 1. Study design and measurement timeline.**

|             | Baseline | Arrival | Briefing | Pre-crisis | Crisis | End sim. | Debrief | Dis-charge |
|-------------|----------|---------|----------|------------|--------|----------|---------|------------|
| CD-RISC 10  | X        |         |          |            |        |          |         |            |
| SDNN (ms)   | X        | X       |          | X          | X      |          | X       | X          |
| PSS-10      | X        |         |          |            |        |          |         |            |
| VAS stress (cm) |      | X       | X        |            |        | X        |         | X          |

*X indicates measurement taken at that time point. Baseline (D–5) = 5 days before simulation. SDNN measured over 3-minute intervals at each time point using a Polar H10 chest strap. VAS stress scored on a 0–10 cm scale.*

### Resilience assessment

Resilience was assessed using the CD-RISC 10 (Connor–Davidson Resilience Scale) [23,24], administered before the simulation (Day –5). This 10-item scale explores the following domains: resilience, emotional and cognitive control, ability to bounce back, sense of self-efficacy, adaptability, and optimism. Each item is rated on a 5-point Likert scale (0–4), with total scores ranging from 0 to 40. Higher scores indicate greater resilience. A CD-RISC 10 score of 26 or below may be considered an indicator of the risk of developing psychosocial problems. [25,26] The French versions of the CD-RISC 10 have demonstrated good psychometric properties with satisfactory internal consistency (α = 0.86) and a confirmed unidimensional structure. [23,24] Use of the scale was authorized by the original authors.

### Physiological stress response

Physiological stress was measured during the simulation using heart rate variability (standard deviation of normal-to-normal intervals, SDNN) [27] recorded with a Polar H10 chest strap and Cardiomood® software. [28] An acute stress situation triggers a sympathetic response leading to decreased heart rate variability. Only time-domain analysis of heart rate variability was performed. Frequency-domain analysis was not conducted, as the 3-minute recording durations per phase were insufficient for reliable spectral analysis in a simulation context involving physical activity. SDNN measurements [4] were obtained for each participant at the following time points: Day –5 (resting measurement, 3 min), upon arrival for the simulation session on Day 0 (3 min), during the briefing (3 min), at the start of the simulation (3 min), during the laryngospasm (3 min), during the debriefing (3 min), and after the debriefing (3 min).

### Perceived stress over the past month

Perceived stress over the past month was assessed at Day –5 using the validated French version of the Perceived Stress Scale 10 (PSS-10). [29] This scale assesses the frequency of situations perceived as threatening. The French version validated by Lesage et al. [29] consists of 10 items rated on a scale of 1–5 (items 4, 5, 7, and 8 are reverse-scored). To allow comparison with the international literature using the original 0–4 rating scale [30], total scores were converted to the 0–40 scale by subtracting 10 from the raw score. Higher scores indicate higher perceived stress levels. The French version demonstrated good internal consistency (α = 0.83). [29]

### Acute psychological stress response

Acute psychological stress was assessed during the simulation using a visual analog scale (VAS) of perceived stress. [31,32] The scale ranges from 0 to 10 cm. Measurements were taken upon arrival for the simulation session, at the start of the simulation after the briefing, immediately after the end of the scenario, during the debriefing, and after the debriefing.

### Simulation protocol

Prior to the simulation, participants received a general briefing on the room layout and the characteristics and limitations of the manikin, including its inability to fully replicate human physiological and emotional responses. Confidentiality, compassion, and the right to make mistakes were emphasized as fundamental principles of simulation-based learning and research. Following completion of the simulation scenario, a structured debriefing was conducted. The instructor team consisted of anesthesiologists and board-certified residents with specific qualifications in medical simulation.

Our standardized, high-fidelity simulation scenario of anesthesia-related pediatric laryngospasm was based on a published scenario [33] and had previously been used in a prior study. [22] It was developed and described in accordance with the guidelines of the French Society for Health Simulation (SOFRASIMS). It involved an anesthesia team of two: an anesthesiology resident and a nurse anesthetist student, or an anesthesiologist and a nurse anesthetist. Working in

pairs is common practice in France, particularly in pediatric anesthesia. Before the scenario began, the team received a one-minute briefing on the current case from the senior anesthesiologist (played by the simulation instructor). After this information transfer, the pair was left alone with the surgeon (played by a neutral embedded facilitator) and the anesthetized patient (SimJunior® manikin; Laerdal Medical, Stavanger, Norway). The primary senior anesthesiologist remained available by phone. After a 3-minute pause, the surgeon performed the skin incision, which was immediately followed by complete laryngospasm with airway obstruction and rapid oxygen desaturation. The scenario ended when the anesthesia team, having resolved the incident, authorized the surgeon to resume the procedure.

Anesthesiology residents or senior anesthesiologists, as well as students or volunteer nurse anesthetists who were unaware of the scenario, were paired to form anesthesia teams. Team composition was determined by availability and training schedules and was not used as an analytical variable. After the simulation, participants provided information via questionnaire regarding their age, years of anesthesia experience since the start of specialized training, prior participation in simulations, pediatric anesthesia experience, and prior exposure to pediatric laryngospasm.

## Statistical analysis

The primary objective was to assess the association between resilience and physiological stress responses during the simulation. Assuming a correlation of 0.5 for a Pearson correlation test against zero correlation and a 5% alpha level, 30 subjects were required to achieve 80% power. Sample size calculation was performed using Stata software version 15.1. Statistical analyses were conducted using R version 4.5.3 (with packages readxl 1.4.5, psych 2.6.3, ggplot2 4.0.2, and rstatix 0.7.3). Qualitative variables were described using counts and percentages. Quantitative variables were described using medians and quartiles. Correlations between two quantitative variables were assessed using Pearson's or Spearman's correlation test, depending on data distribution, with bootstrap confidence intervals (2,000 resamples) for the primary objective. The relationship between a binary variable and a quantitative variable was assessed using Student's t-test or a Wilcoxon test, depending on data distribution. Tests were conducted with a two-sided alpha level of 5%.

Additional exploratory analyses, not specified in the initial protocol, were performed to better characterize the sample and to verify that the simulation induced a measurable stress response. Kruskal–Wallis tests were used to compare the main variables across the four professional groups. The temporal evolution of stress markers (SDNN and VAS) was assessed using the Friedman test. These post-hoc analyses were not preregistered, and their results should be interpreted with caution. No correction for multiple comparisons was applied to the secondary and exploratory analyses.

## Results

Between June and October 2021, 19 volunteer pairs of anesthetists were recruited. Four participants were excluded post hoc: one due to antiarrhythmic medication use identified after inclusion, and three due to unusable physiological data resulting from technical failure of heart rate variability recording during the simulation. Ultimately, 34 subjects were included in the analysis. Table 2 presents the participant characteristics by professional group.

The internal consistency of the CD-RISC 10 in our sample was α = 0.65. The internal consistency of the PSS-10 was α = 0.81.

### Resilience and baseline stress levels

The median [Q1–Q3] baseline resilience score was 29 [27–32], ranging from 18 to 34 (Fig 1: distribution of CD-RISC 10 scores). Eight participants (23.5%) had a score of 26 or below, a threshold associated with increased psychosocial risk. The median perceived stress score over the past month was 15 [13–18], ranging from 2 to 27; the median basal SDNN was 71 [56–81] ms, ranging from 36 to 135; the median baseline acute perceived stress score was 2 [1–4] cm, ranging from 0 to 8.

**Table 2. Participant characteristics by professional group.**

| Characteristic | Overall (n = 34) | Senior Anesth. (n = 9) | Residents (n = 9) | Nurse Anesth. (n = 10) | Student NA (n = 6) |
|---|---|---|---|---|---|
| Age, years | 35 [29 –40] | 41 [35 –48] | 28 [26 –29] | 37 [35 –42] | 32 [32 –37] |
| Experience, years | 4.0 [2.0–9.4] | 17.0 [9.5–25.0] | 2.0 [1.5–4.0] | 6.5 [3.2–8.8] | 2.0 [2.0–2.0] |
| Laryngospasm exp., n/N | 20/33 (60.6%) | | | | |
| CD-RISC 10 score | 29 [27 –32] | 29 [28 –32] | 28 [27 –31] | 29 [28 –32] | 26 [24 –28] |
| PSS-10 score | 15 [13 –18] | 13 [10 –18] | 14 [13 –16] | 16 [14 –20] | 16 [11 –22] |
| Basal SDNN, ms | 71.2 [56.7–81.6] | 76.6 [68.0–92.9] | 71.7 [70.7–84.5] | 61.7 [52.0–74.0] | 60.6 [53.5–73.2] |
| Baseline VAS stress, cm | 2.0 [1.0–4.0] | 2.0 [1.0–3.0] | 1.0 [1.0–2.0] | 4.0 [2.2–5.0] | 5.0 [1.0–6.8] |

Values are median [Q1–Q3] unless otherwise specified. NA = nurse anesthetist.

CD-RISC 10: Connor–Davidson Resilience Scale; PSS-10: Perceived Stress Scale; SDNN: standard deviation of normal-to-normal intervals; VAS: visual analog scale.

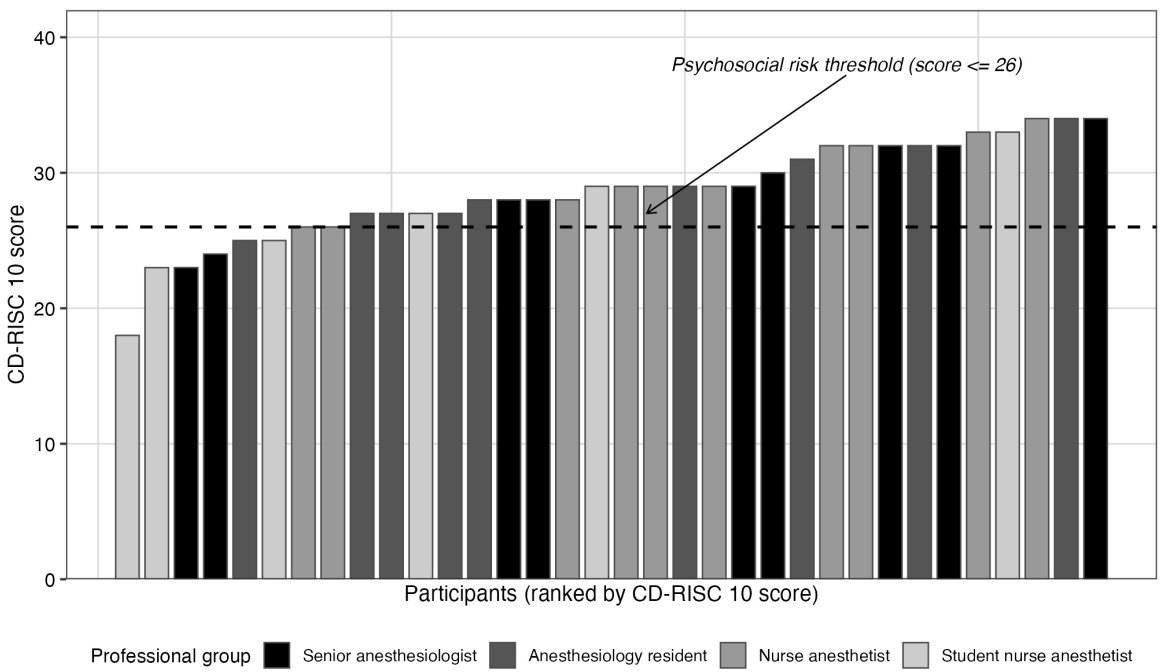

**Fig 1. Distribution of resilience scores.** Distribution of CD-RISC 10 scores among anesthesia professionals. The horizontal line represents the threshold below which psychosocial risk is higher.

## Primary objective: Association between resilience (CD-RISC 10) and physiological stress response (H1)

Regarding the primary hypothesis (H1), Spearman's correlation analysis between CD-RISC 10 scores and SDNN during laryngospasm revealed no statistically significant association (Spearman's rho = −0.299, 95% bootstrap CI [−0.653; +0.134], p = 0.086). Fig 2 illustrates the relationship between resilience and SDNN during the crisis phase.

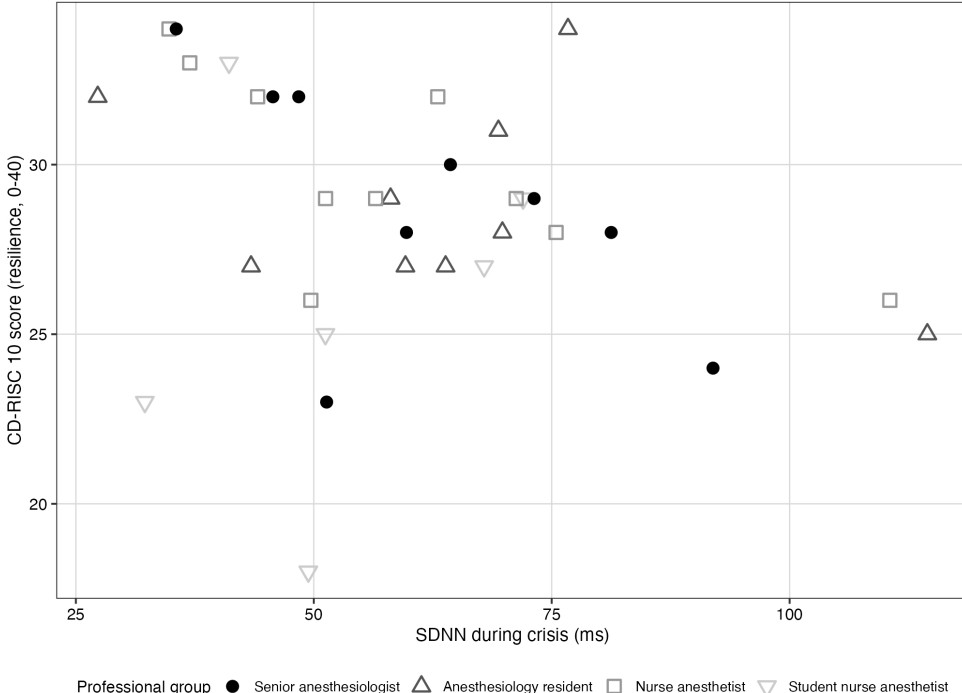

Professional group ● Senior anesthesiologist △ Anesthesiology resident □ Nurse anesthetist ▽ Student nurse anesthetist

**Fig 2. Relationship between resilience (CD-RISC 10) and physiological stress during laryngospasm (SDNN, ms).** Each point represents one participant; shapes and shading indicate professional group.

### Secondary objectives (H2 to H4)

Regarding hypothesis H2, no association was found between resilience and baseline acute perceived stress (rho = −0.088, p = 0.621). An isolated association was found with the post-simulation VAS (rho = −0.416, p = 0.015), which was not retained after considering the multiplicity of tests.

Regarding H3, no association was found between resilience and perceived stress over the past month (rho = −0.256, p = 0.144). Fig 3 illustrates this relationship.

Hypothesis H4 was not supported. No association was found between resilience and age (rho = −0.061, p = 0.733) or between resilience and experience (rho = +0.222, p = 0.207).

The correlation results with the CD-RISC 10 are presented in Table 3.

### Post-hoc exploratory analysis

Comparison between professional groups using the Kruskal–Wallis test showed no significant differences for the CD-RISC 10 (H(3) = 3.166, p = 0.367), PSS-10 (H(3) = 1.921, p = 0.589), SDNN during laryngospasm (H(3) = 1.572, p = 0.666), or post-simulation VAS (H(3) = 0.956, p = 0.812).

The temporal evolution of acute stress markers (SDNN and VAS stress) was analyzed using the Friedman test. Both SDNN and VAS showed significant temporal variation during the simulation (Friedman test, $\chi^2(5) = 24.03$, p < 0.001 and $\chi^2(4) = 53.87$, p < 0.0001, respectively), confirming that the simulation induced a measurable stress response. Fig 4 illustrates this temporal variation for acute stress markers.

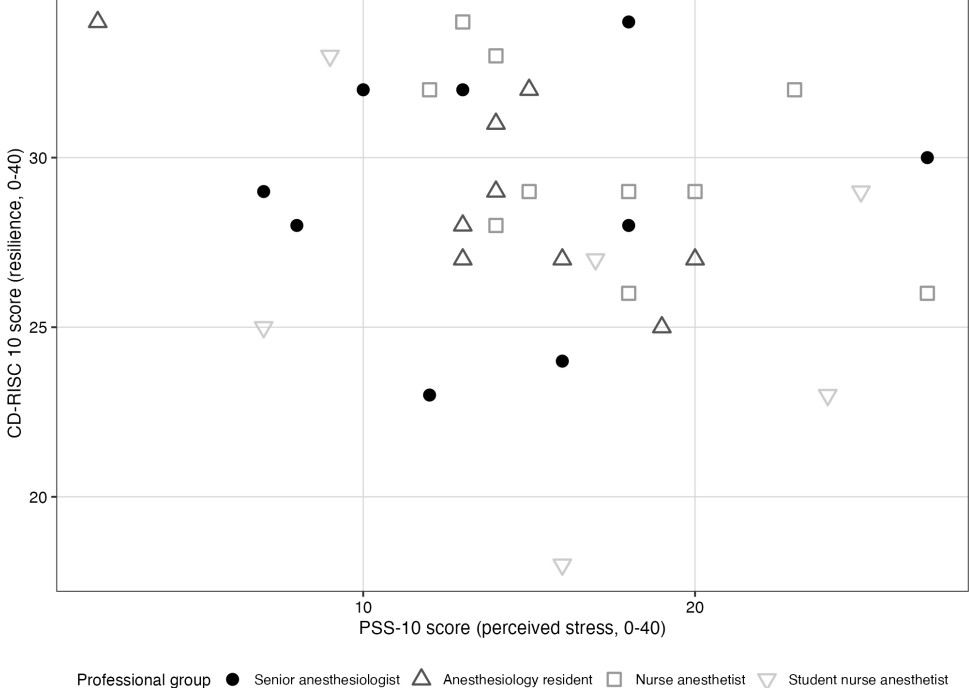

**Fig 3. Relationship between resilience (CD-RISC 10) and perceived stress over the past month (PSS-10).** Scatter plot illustrating individual resilience (CD-RISC 10) and perceived stress (PSS-10) scores among participants. Each point represents one participant's scores on both scales.

**Table 3. Spearman correlations between resilience (CD-RISC 10) and stress measures.**

| Parameter | Spearman rho | Bootstrap 95% CI | p-value | n |
|---|---|---|---|---|
| **SDNN during laryngospasm, ms** | **−0.299** | **[−0.653; +0.134]** | **0.086** | **34** |
| Basal SDNN, ms | −0.152 | [−0.457; +0.212] | 0.391 | 34 |
| PSS-10 | −0.256 | [−0.561; +0.076] | 0.144 | 34 |
| Baseline VAS stress | −0.088 | [−0.428; +0.263] | 0.621 | 34 |
| VAS stress pre-briefing | −0.170 | [−0.520; +0.199] | 0.337 | 34 |
| VAS stress post-briefing | −0.094 | [−0.440; +0.288] | 0.598 | 34 |
| VAS stress post-simulation | −0.416 | [−0.725; −0.035] | 0.015* | 34 |
| VAS stress post-debriefing | −0.122 | [−0.421; +0.215] | 0.491 | 34 |
| Age, years | −0.061 | | 0.733 | 34 |
| Experience, years | +0.222 | | 0.207 | 34 |

*Primary outcome in bold. Bootstrap 95% CI computed with 2,000 resamples. * p < 0.05 (uncorrected). No correction for multiple comparisons was applied.*

## Discussion

In this prospective study involving 34 anesthesia professionals exposed to a high-fidelity simulation of pediatric laryngospasm, we found no association between resilience and acute stress (physiological and perceived), perceived stress over the past month, age, or experience.

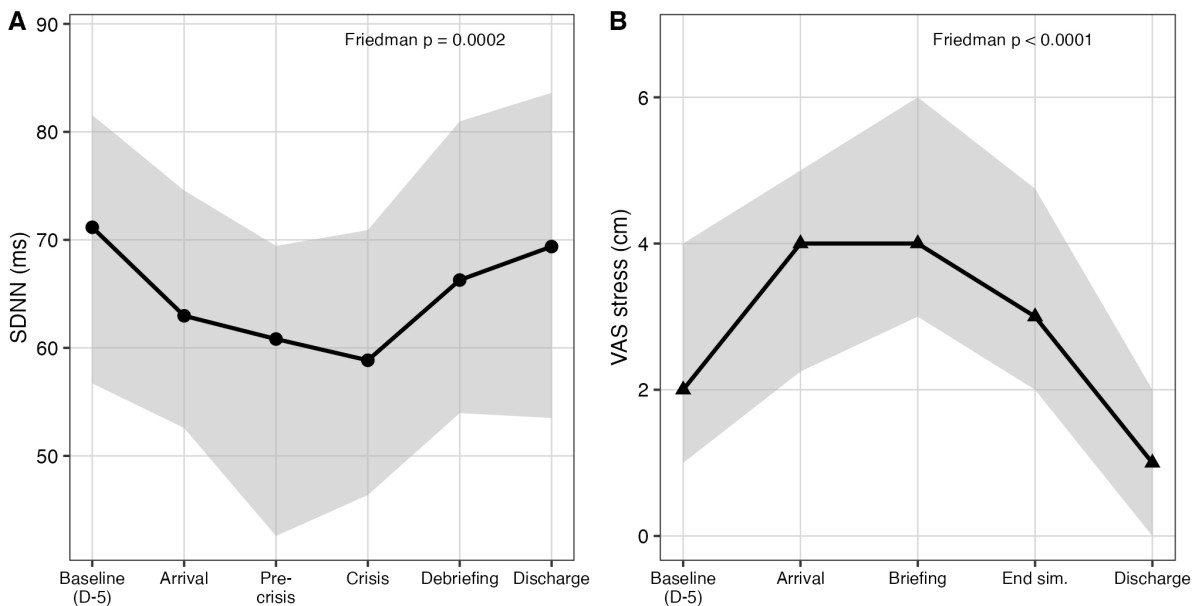

**Fig 4. Temporal variation for acute stress marker (SDNN and VAS Stress).**

This lack of significant findings regarding resilience and the acute stress response may be explained by the fact that resilience manifests over time as a process of adaptation to adversity, rather than as a personality trait that predicts an immediate response to acute stress. [19,34] According to Lazarus and Folkman, the stress response depends on the individual's cognitive appraisal of the situation based on available resources. [1] Resilience modulates this appraisal but does not necessarily attenuate the immediate physiological response, which is largely automatic. The relationship between physiological stress markers and self-reported measures is complex and depends on numerous individual factors. [4] SDNN reflects autonomic activation, which exhibits substantial inter-individual variability; this may explain the absence of a direct correlation with resilience.

The observed median resilience score (29 [27–32]) was comparable to that of other studies involving anesthesia professionals but lower than the score reported in the general U.S. population (32.1±5.8). [25,35] The absence of an association with perceived stress over the past month may be due to an overestimated hypothesized correlation of 0.5. Indeed, a 2015 study by Rushton et al. found an association between resilience and perceived stress of −0.44. [20] Eight participants (23.5%) had a CD-RISC 10 score of 26 or below, a threshold below which increased psychosocial risk has been suggested. [25,26] This proportion is comparable to that reported by Douillet et al. among French physicians during the COVID-19 pandemic. [25]

A significant isolated association was observed between resilience and perceived stress immediately following the simulation (rho = −0.416, p = 0.015). Although this may suggest that greater resilience is associated with lower perceived stress following an acute event, this finding must be interpreted with great caution. Given the multiple secondary comparisons performed without correction, this association may be a chance finding. Conversely, if confirmed, it could reflect the influence of resilience on the immediate cognitive appraisal of a past stressor. This hypothesis warrants testing in future studies with adequate statistical power and predefined analyses.

Regarding our study protocol, the post-hoc analysis confirms that the simulation induced measurable stress. Acute anticipatory stress (SDNN and VAS pre-briefing) is consistent with the literature on simulation-induced stress. [21,36]

## Study strengths

First, our study protocol was preregistered before the start of recruitment, in accordance with current recommendations. The simulations were multi-professional, immersive, and realistic. The VAS stress score at the end of the simulation was low, suggesting that the debriefing was effective and that the psychological safety of participants was maintained. To our knowledge, this is the first study to attempt to measure the interactions between stress and resilience across the full range of anesthesia professions (staff anesthesiologist, anesthesiology resident, nurse anesthetist, and nurse anesthetist student).

## Limitations

There are, however, several methodological limitations. Given the sample size, we may not have been able to detect an association between resilience and stress. The study was conducted at a single site, which limits the generalizability of the results. Since all participants volunteered for the simulation sessions, we cannot assert that the results would be identical among those who declined to participate. The scales used are self-report measures, and although they are all validated, we cannot rule out the possibility that participants did not objectively report their feelings at the time. The internal consistency of the CD-RISC 10 in our sample was $\alpha = 0.65$, which is lower than published validation values. This may be due to the small sample size and possibly the heterogeneity of our population. Data on sex and gender were not collected, which constitutes a limitation. Finally, the study did not compare results across high- and low-stress scenarios, due to logistical constraints and limited participant availability.

Looking ahead, it appears necessary to conduct longitudinal studies with long-term follow-up to characterize resilience over time as a dynamic process. Sample sizes in such studies should be larger. It would also be worthwhile to explore these relationships in other cultural contexts and healthcare systems, particularly in resource-limited settings where working conditions and structural factors may influence psychological adaptation processes differently. Such studies could help describe mediation or moderation models between resilience and stress. Finally, since stress affects clinical performance levels, it would be of interest to examine whether a mediating relationship exists between resilience and performance.

## Conclusion

We found no association between resilience and the various measures of acute-phase stress. Consequently, the assessment of the acute-phase stress response does not appear to accurately reflect the level of resilience among anesthesia professionals. Predicting exposure to stress-induced psychosocial risks remains challenging among anesthesia professionals. Directly measuring resilience could serve as a complementary tool for identifying professionals at psychosocial risk, rather than relying solely on observing their reactions in acute stress situations.

## Supporting information

**S1 Data. Dataset RESISTRESS study: resilience and stress measures in anesthesia professionals.**
(XLSX)

## Acknowledgments

An AI language model (Claude, Anthropic) was used to improve the English translation of the manuscript. This AI tool was used solely for language translation and manuscript formatting. All scientific content, data analysis, interpretation, and intellectual contributions are the sole responsibility of the authors. All authors reviewed, edited, and approved the final English version of the manuscript.

## Author contributions

**Conceptualization:** Yoann Zafiriou, Jean-Noël Evain, Guillaume Archer, Joris Botton, Julien Crozet, Ismaël Khediri, Mathieu Lefevre, Camille Marcel, Charline Sery, Anne Ego, Pierre Albaladejo, Julien Picard.

**Data curation:** Yoann Zafiriou, Jean-Noël Evain, Guillaume Archer, Joris Botton, Julien Crozet, Ismaël Khediri, Mathieu Lefevre, Camille Marcel, Charline Sery.

**Formal analysis:** Yoann Zafiriou, Jean-Noël Evain, Guillaume Archer, Joris Botton, Julien Crozet, Ismaël Khediri, Mathieu Lefevre, Camille Marcel, Charline Sery, Anne Ego, Pierre Albaladejo, Julien Picard.

**Investigation:** Yoann Zafiriou, Jean-Noël Evain, Barthélémy Bertrand, Guillaume Archer, Joris Botton, Julien Crozet, Ismaël Khediri, Mathieu Lefevre, Camille Marcel, Charline Sery.

**Methodology:** Yoann Zafiriou, Jean-Noël Evain, Barthélémy Bertrand, Guillaume Archer, Joris Botton, Julien Crozet, Ismaël Khediri, Mathieu Lefevre, Camille Marcel, Charline Sery, Anne Ego, Pierre Albaladejo, Julien Picard.

**Supervision:** Yoann Zafiriou, Jean-Noël Evain, Anne Ego, Pierre Albaladejo, Julien Picard.

**Visualization:** Yoann Zafiriou.

**Writing – original draft:** Yoann Zafiriou, Jean-Noël Evain, Pierre Albaladejo, Julien Picard.

**Writing – review & editing:** Yoann Zafiriou, Jean-Noël Evain, Barthélémy Bertrand, Guillaume Archer, Joris Botton, Julien Crozet, Ismaël Khediri, Mathieu Lefevre, Camille Marcel, Charline Sery, Anne Ego, Pierre Albaladejo, Julien Picard.

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
