## [Decision Letter · Decision Letter 0]

31 Mar 2026

PONE-D-26-06857

How is stress related to resilience in anesthesia professionals? A simulation-based prospective study.

PLOS One

Dear Dr. Zafiriou,

Thank you for submitting your manuscript to PLOS ONE. After careful consideration, we feel that it has merit but does not fully meet PLOS ONE’s publication criteria as it currently stands. Therefore, we invite you to submit a revised version of the manuscript that addresses the points raised during the review process.

As the corresponding author, your ORCID iD is verified in the submission system and will appear in the published article. PLOS supports the use of ORCID, and we encourage all coauthors to register for an ORCID iD and use it as well. Please encourage your coauthors to verify their ORCID iD within the submission system before final acceptance, as unverified ORCID iDs will not appear in the published article. Only the individual author can complete the verification step; PLOS staff cannot verify ORCID iDs on behalf of authors.

We look forward to receiving your revised manuscript.

Kind regards,

Ilse Bloom

Staff Editor

PLOS One

Journal Requirements:

2. Please ensure that you refer to Figures 2 and 3 in your text as, if accepted, production will need this reference to link the reader to the figure.

3. We note that there is identifying data in the Supporting Information file <S1 dataset resistress.xls> Due to the inclusion of these potentially identifying data, we have removed this file from your file inventory. Prior to sharing human research participant data, authors should consult with an ethics committee to ensure data are shared in accordance with participant consent and all applicable local laws.

-Location data

Please remove or anonymize all personal information (Age), ensure that the data shared are in accordance with participant consent, and re-upload a fully anonymized data set. Please note that spreadsheet columns with personal information must be removed and not hidden as all hidden columns will appear in the published file.

**Additional Editor Comments:**

The manuscript has been evaluated by two reviewers, and their comments are available below.

Could you please carefully revise the manuscript to address all comments raised?

Reviewers' comments:

Reviewer's Responses to Questions

**Comments to the Author**

1. Is the manuscript technically sound, and do the data support the conclusions?

Reviewer #1: Partly

Reviewer #2: Partly

2. Has the statistical analysis been performed appropriately and rigorously? 

Reviewer #1: No

Reviewer #2: I Don't Know

3. Have the authors made all data underlying the findings in their manuscript fully available?

Reviewer #1: No

Reviewer #2: Yes

4. Is the manuscript presented in an intelligible fashion and written in standard English?

Reviewer #1: No

Reviewer #2: Yes

5. Review Comments to the Author

Reviewer #1: Dear Authors, on behalf of the journal PLOS ONE, we thank you for submitting your manuscript titled “How is stress related to resilience in anesthesia professionals? A simulation-based prospective study.” The submitted manuscript is original; however, we suggest reducing the similarity percentage, which according to Turnitin software is 36%, and we note that 0% of the text was generated by AI. We suggest improving the following aspects: 1) Improve the wording of the study objective, which should be consistent with the study title. The abstract should state a clear and precise objective (line 30). 2) Regarding the instrument used to measure chronic stress, the PSS-10 is not appropriate, as it is a screening test that measures perceived stress. To measure chronic stress in professionals, there are other validated tests such as Maslach’s MBI and others; it is suggested to reformulate this section and to clarify this point in the introduction and secondary objectives (lines 65–68). 3) Improve the wording of the methods section. It is suggested to write in an orderly manner with logical coherence. This is an observational-analytical study that seeks to establish an association between stress and resilience variables. It is necessary to specify the unit of analysis, population, sampling method, sample size, and inclusion criteria. Additionally, the methodology and the validity and reliability of the instruments used should be described. This should be supported by bibliographic references regarding the measurement of physiological stress and psychological stress. It is also not specified whether the psychological tests were used with permission, were freely available, or were purchased. Improve the wording regarding the administration of the instruments; be clear and precise. 4) In the data analysis section, specify the statistical methods used for descriptive and correlational analysis. 5) Improve the presentation of results in accordance with the study’s objectives. An initial table of sample characteristics. 6) The discussion states that the sample is representative but does not specify the sampling method; review this aspect and improve the wording. The results obtained are consistent with contemporary theoretical frameworks, which conceptualize resilience not as an immediate reaction to stress, but as a dynamic process of adaptation that manifests over time and in interaction with multiple contextual and individual factors. It is suggested to compare the results with theories or explanatory models of the variables studied and previous literature from the last 5 years. It is suggested that the discussion include suggestions for future research, which should consider longitudinal designs and more complex analytical models, such as mediation and moderation analyses, to explore the interaction between stress, resilience, and other psychosocial factors in clinical contexts. Likewise, it would be relevant to examine these relationships in different cultural contexts and health systems, particularly in settings with limited resources, where work demands and structural factors can have varying effects on psychological adaptation processes. 7) The conclusion should be consistent with the study's objective. We suggest improving the wording.8)The attached dataset is incomplete; only the initial data has been included, and the final data is missing.

Reviewer #2: This is a simulation-training based study aiming to examine the relationship between stress and resiliency in anaesthesia providers.

It is assumed that the high psychological fidelity of the simulator setting creates realistic psychological challenges and thus, analogous to real life emergency situations, simulation-based training produces a significant stress response. The authors are to be commended for using the simulation lab not only for training but also for research in order to contribute to the stress-performance-personality literature in anesthesia.

Overall he paper is concise and reasonably well written. However there are a few important points which need to be addressed in order to provide this promising and original paper with sufficient depth.

Introduction:

In the introduction terms such as stress, chronic stress, occupational stress, physiological stress response, psychological trauma and also burnout, resilience and stress management are all used. They are not interchangeable. Psychological concepts such as stress and resiliency need to be clearly defined. I would like to suggest the authors to read this paper: An Overly Permissive Extension. Jerome Kagans. Perspectives on Psychological Science 2016, Vol. 11(4) 442–450.

Furthermore, hypothesized associations between personality characteristics such as resiliency and subsequent stress responses (acute/chronic and physiological/psychological) should be explained within a theoretical framework of stress, eg the transactional stress model (Lazarus and Folmon) or the allostatic load model (McEwan). When not clearly defined and embedded in theory, concepts loose meaning and associations become insignificant.

Please rewrite this entire section. Both a conceptual framework and hypothesis are lacking.

-What do you mean in this paper when discussing stress?

-Why is it important for anesthesiologists and their practice?

-Why do you expect an association between physiological stress response, a self-reporting stress and self-reported resiliency? What does that mean?

Methodology:

-The study was preregistered: very good.

-Why didn’t you choose to differ the experimental condition will differ within the participant? Ie. confronting each participant with two scenarios: one of which will be a low stress scenario and the other will be a high stress scenario.

-Please provide psychometric properties of questionnaire instruments used.

-Did you use frequency domain analysis of HRV? Please elaborate.

Results:

When reporting results of questionnaires often the psychometric properties, internal consistencies, are reported. Why didn’t you do that in your paper?

-165: You report that no correction for multiple comparisons was applied. You explain because it is an exploratory study: why not hypothesize? Ther is more than enough background if you rewrite you introduction.

-And lines 179: No statistically significant associations were observed between resilience and physiological or perceived stress measures. An isolated association observed in exploratory analyses was not retained after considering the multiplicity of tests. See above: why not hypothesize.

Discussion

-Dissociations between biomarker measures of stress and self-reported measures of stress and characteristics of personality are frequently reported in the literature. This paper would be significantly more interesting when discussing why you also found this dissociation, while according to theoretical background associations between stress conditions-acute and chronic stress should be hypothesized.

-216 yes, stress response is important in terms of performance. Is this relevant for your paper? Then please elaborate why here and elsewhere. Performance is not measured in this study. Otherwise keep to the theme the relationship between acute and chronic stress responses, physiological and psychological and it’s importance for well-being.

-223 and further. The line of reasoning in this paragraph is confusing. This paragraph seems to be written with an ill understanding of the subject of stress. Concepts are insufficiently defined and connected ad hoc. What do you mean here? This is undermining the paper. Please rewrite or omit.

6. PLOS authors have the option to publish the peer review history of their article (what does this mean?). If published, this will include your full peer review and any attached files.

Reviewer #1: No

Reviewer #2: No

---

## [Author Response · Author response to Decision Letter 1]

10 Apr 2026

We thank the editor and both reviewers for their thorough and constructive evaluation. A detailed point-by-point response to all comments is provided in the attached "Response to Reviewers" document. The manuscript has been substantially revised, including a rewritten Introduction with a theoretical framework and explicit hypotheses, restructured Methods and Results sections, a fully anonymized dataset, and new analyses as requested.

---

## [Decision Letter · Decision Letter 1]

12 May 2026

Is resilience linked to stress response among anesthesia professionals? A prospective simulation-based study.

PONE-D-26-06857R1

Dear Dr. Zafiriou,

We’re pleased to inform you that your manuscript has been judged scientifically suitable for publication and will be formally accepted for publication once it meets all outstanding technical requirements.

Kind regards,

Siraj Ahmed Ali

Academic Editor

PLOS One

Additional Editor Comments (optional):

Reviewers' comments:

Reviewer's Responses to Questions

**Comments to the Author**

1. If the authors have adequately addressed your comments raised in a previous round of review and you feel that this manuscript is now acceptable for publication, you may indicate that here to bypass the “Comments to the Author” section, enter your conflict of interest statement in the “Confidential to Editor” section, and submit your "Accept" recommendation.

Reviewer #1: All comments have been addressed

Reviewer #2: All comments have been addressed

2. Is the manuscript technically sound, and do the data support the conclusions?

Reviewer #1: Yes

Reviewer #2: Yes

3. Has the statistical analysis been performed appropriately and rigorously? 

Reviewer #1: Yes

Reviewer #2: Yes

4. Have the authors made all data underlying the findings in their manuscript fully available?

Reviewer #1: Yes

Reviewer #2: Yes

5. Is the manuscript presented in an intelligible fashion and written in standard English?

Reviewer #1: Yes

Reviewer #2: Yes

6. Review Comments to the Author

Reviewer #1: Warm regards to the authors, and I would like to commend your efforts to address the comments made on the manuscript submitted to PLOS ONE. I have reviewed the manuscript and the comments have been addressed. However, I would suggest implementing just two further improvements: the first four lines of the abstract could be included in the ‘Background’ section, with the remainder in the ‘Objectives’ section. In the ‘Methods’ section, it would be helpful to specify the period during which the study was conducted.

Reviewer #2: Dear authors. I would like to compliment you with this thorough revision. My comments (#2) have been adequately and conscientiously addressed. I have no further critique and wish you all the best. Regards, RW.

7. PLOS authors have the option to publish the peer review history of their article (what does this mean?). If published, this will include your full peer review and any attached files.

Reviewer #1: No

Reviewer #2: No

---

## [Editor Report · Acceptance letter]

PONE-D-26-06857R1

PLOS One

Dear Dr. Zafiriou,

I'm pleased to inform you that your manuscript has been deemed suitable for publication in PLOS One. Congratulations! Your manuscript is now being handed over to our production team.

Kind regards,

on behalf of

Dr. Siraj Ahmed Ali

Academic Editor

PLOS One